# Apelin as a Potential Regulator of Peak Athletic Performance

**DOI:** 10.3390/ijms24098195

**Published:** 2023-05-03

**Authors:** Roland Ligetvári, István Szokodi, Gabriella Far, Éva Csöndör, Ákos Móra, Zsolt Komka, Miklós Tóth, András Oláh, Pongrác Ács

**Affiliations:** 1Doctoral School of Health Sciences, University of Pécs, 7621 Pécs, Hungary; 2Faculty of Health Sciences, University of Pécs, 7621 Pécs, Hungary; 3Heart Institute, Medical School, University of Pécs, 7624 Pécs, Hungary; 4Szentágothai Research Centre, University of Pécs, 7624 Pécs, Hungary; 5Department of Laboratory Medicine, Semmelweis University, 1085 Budapest, Hungary; 6Department of Health Sciences and Sport Medicine, Hungarian Sports University, 1123 Budapest, Hungary; 7Heart and Vascular Center, Semmelweis University, 1122 Budapest, Hungary

**Keywords:** apelin, peak performance, myokine, exercise, differential response, VO_2_max

## Abstract

Apelin, as a cardiokine/myokine, is emerging as an important regulator of cardiac and skeletal muscle homeostasis. Loss of apelin signaling results in premature cardiac aging and sarcopenia. However, the contribution of apelin to peak athletic performance remains largely elusive. In this paper, we assessed the impact of maximal cardiorespiratory exercise testing on the plasma apelin levels of 58 male professional soccer players. Circulating apelin-13 and apelin-36, on average, increased transiently after a single bout of treadmill exercise; however, apelin responses (Δapelin = peak − baseline values) showed a striking interindividual variability. Baseline apelin-13 levels were inversely correlated with those of Δapelin-13 and Δapelin-36. Δapelin-13 showed a positive correlation with the maximal metabolic equivalent, relative maximal O_2_ consumption, and peak circulatory power, whereas such an association in the case of Δapelin-36 could not be detected. In conclusion, we observed a pronounced individual-to-individual variation in exercise-induced changes in the plasma levels of apelin-13 and apelin-36. Since changes in plasma apelin-13 levels correlated with the indicators of physical performance, whole-body oxygen consumption and pumping capability of the heart, apelin, as a novel exerkine, may be a determinant of peak athletic performance.

## 1. Introduction

Peak performance, especially peak athletic performance, and the accomplishment of becoming the strongest, the fastest, and the fittest has been the goal of athletes for centuries. Several factors play a role in maximizing performance: genetic background, mental toughness, training and cardiorespiratory fitness, among others [1,2,3,4,5]. The cardiovascular system is the engine of cardiorespiratory fitness and has been the focus of research in exercise physiology and sports medicine for decades. Research about exercise adaptation dates back to more than 50 years [6] and has become more common in recent decades with different acute exercise models [7,8,9,10,11]. 

Reaching the maximum of exercise capacity is a major hemodynamic challenge for the cardiovascular system. Adjusting parameters to provide a sufficient blood supply to meet the increased O_2_ demand of the skeletal muscles is a highly coordinated effort of the O_2_ delivery and O_2_ extraction systems [12,13]. Exerkines represent a broad variety of signaling molecules released in response to exercise that exert their effects through endocrine, paracrine, and/or autocrine pathways. A multitude of organs and tissues release these factors: cardiokines, myokines, and adipokines are produced and secreted from the heart, skeletal muscle, and white adipose tissue, respectively. Exerkines are emerging as crucial mediators of exercise adaptation via their role in local regulation, interorgan and systemic cross-talk [14,15,16].

Apelin is an endogenous ligand of the G-protein-coupled apelin receptor, originally isolated from bovine stomach extracts as a 77-amino-acid prepropeptide. The prepro-apelin is further cleaved into shorter but functional fragments, including apelin-36, apelin-17, and apelin-13 [17], with the shorter fragments showing higher affinity for the apelin receptor [18]. Since its discovery, apelin has been described as a cardiokine [19], myokine [20], as well as an adipokine [21]. 

The apelin–apelin receptor system is a potent regulator of the cardiovascular system [18,22,23]. In pre-clinical models, the peptide stimulates cardiac contractility [19,24], induces vasodilatation in various vascular beds in vitro [25,26,27], and elicits a blood-pressure-lowering effect in vivo [17,28]. While the vasorelaxant and hypotensive effects of apelin are nitric-oxide-dependent [17,25,26,27,28], the positive inotropic effect of the peptide is not blunted by nitric oxide synthase inhibition [19]. In line with the hemodynamic changes observed in experimental conditions, the acute administration of apelin in humans causes peripheral and coronary vasodilatation and increases cardiac output [29]. Of importance, prolonged endurance exercise training upregulates skeletal muscle apelin expression [20], and skeletal muscle cell contraction stimulates apelin secretion [30]. The skeletal-muscle-specific knockdown of apelin results in muscle weakness, reduced exercise capacity, and impaired exercise-induced elevation of plasma apelin levels in mice. These beneficial effects are related, in part, to the powerful impact of apelin on the metabolism. Apelin treatment promotes a systemic glucose-lowering effect associated with enhanced glucose utilization in white adipose tissue and skeletal muscle [31]. Moreover, chronic apelin administration increases skeletal muscle mitochondrial function and biogenesis [30]. Collectively, apelin is a strong candidate for participating in cardiac and skeletal muscle adaptation to physical exercise. Several studies have described the effect of physical exercise on apelin levels [7,9,32,33]; however, the contribution of different apelin isoforms to peak athletic performance in professional soccer players has not been characterized. Soccer is a team sport, where individual players need a combination of endurance, strength, sprinting, and jumping skills for shorter time periods, meaning that aerobic and anaerobic demands are both present [10].

For these reasons, our aims in the present study are to examine the acute changes in apelin-13 plasma levels following a single bout of maximum exercise in professional soccer players and the possible association between apelin-13 and metabolic- and cardiopulmonary-exercise-derived parameters. Additionally, we aim to characterize the association between apelin-13 and apelin-36 in a subpopulation of athletes. 

## 2. Results

### 2.1. Response to a Single Bout of Exercise

A total of 58 athletes participated in the vita maxima treadmill test. The subjects run until maximum exhaustion, which lasted for 11.2 ± 1.5 min (minimum: 7.3 min; maximum: 15.2 min). Cardiovascular, cardiorespiratory, and metabolic parameters were recorded for all participants. Blood pressure (BP), heart rate (HR), and lactate concentration was recorded at rest (baseline), at maximum load (peak), and 30 min after the maximum load (recovery) (Table 1).

The repeated measures one-way analysis of variance (ANOVA) revealed a significant effect of exercise intervention on the diastolic blood pressure (DBP) (F (1.97, 132) = 36.9; *p* < 0.001). Peak DBP decreased significantly in recovery (Tukey’s multiple comparisons test, *p* < 0.001), and recovery DBP was significantly lower than baseline DBP (Tukey’s multiple comparisons test, *p* < 0.001).

The Friedman test revealed a significant effect of exercise intervention on systolic blood pressure (Friedman statistic: 127, *p* < 0.001), heart rate (Friedman statistic: 125, *p* < 0.001), and blood lactate (Friedman statistic: 140, *p* < 0.001). Systolic blood pressure (SBP) increased at peak load compared to baseline and decreased in recovery compared to peak and baseline (Dunn’s multiple comparisons test, *p* < 0.001 for all three comparisons). Heart rate increased from baseline to peak load and, in recovery, it decreased to a level lower than peak but higher than baseline (Dunn’s multiple comparisons test, *p* < 0.001 for all three comparisons). Blood lactate concentration increased at peak load compared to baseline and decreased in recovery compared to peak (Dunn’s multiple comparisons test, *p* < 0.001 for all three comparisons).

An average maximum O_2_ consumption (VO_2_max) of 3.97 ± 0.57 L/min (median (IQR) = 3.95 (3.65–4.32)) and relative VO_2_max of 52.7 (47–57.3) mL/kg/min (median (IQR)) was achieved at maximum load. We found positive associations between relative VO_2_max and maximum metabolic equivalent (MET) (Spearman correlation; r = 0.975, *p* < 0.001), maximum CO_2_ production (VCO_2_) (Spearman correlation; r = 0.467, *p* < 0.001), circulatory power (Spearman correlation; r = 0.847, *p* < 0.001), and circulatory stroke work (Spearman correlation; r = 0.785, *p* < 0.001), and negative associations between relative VO_2_max and maximum respiratory quotient (RQ) (Spearman correlation; r = −0.561, *p* < 0.001), maximum ventilation/maximum O_2_ consumption (VE/VO_2_) (Spearman correlation; r = −0.505, *p* < 0.001), and VE/VCO_2_ (Spearman correlation; r = −0.339, *p* = 0.005).

The level of N-terminal pro-B-type natriuretic peptide (NT-proBNP) did not change during or after the treadmill test (baseline: 41.4 (20.6–74.5) pmol/L; peak: 35.2 (21.4–67.5) pmol/L; recovery: 43.1 (30.9–66.4) pmol/L; Friedman statistic: 2.44, *p* = 0.296).

### 2.2. Apelin-13 Response to a Single Bout of Exercise

Peptide concentrations were measured at baseline, peak, and recovery (Figure 1). The baseline apelin-13 level was 144 ± 72.4 pg/mL (95% CI, 124–163 pg/mL), peak apelin-13 was 167 ± 71.5 pg/mL (95% CI, 147–186 pg/mL), and recovery apelin-13 was 137 ± 65.7 pg/mL (95% CI, 119–155 pg/mL). The repeated measures one-way ANOVA revealed a significant effect of the exercise intervention on apelin-13 levels (F (1.75, 92.8) = 6.76; *p* = 0.003). Tukey’s multiple comparisons test showed a difference between baseline vs peak, and peak vs recovery values (Figure 1A,B).

We found a negative correlation between the peak–baseline values for apelin-13 (Δapelin-13) and baseline apelin-13 (Figure 1C). As the general response to the acute load was an increase in apelin-13, we analyzed the peptide response on an individual level. Figure 1D shows the Δapelin-13 for all soccer players in ascending order. The response to physical load was heterogeneous, with a mean Δapelin-13 level of 21.2 ± 63.5 pg/mL.

### 2.3. Correlation of Apelin-13 with Cardiopulmonary-Exercise-Derived Parameters

We found a positive correlation between Δapelin-13 and baseline blood pressure, peak diastolic blood pressure, circulatory power, maximum MET, and relative VO_2_max (Figure 2).

The players were divided into two groups using the median value of Δapelin-13 (Table 1). Baseline SBP and DBP were significantly lower in the low responder group than in the high responder group (SBP: 139 ± 12 mm Hg vs 147 ± 13 mm Hg, *p* = 0.048; and DBP 77 ± 8 mm Hg vs. 85 ± 7 mm Hg, *p* = 0.001). 

In the low responder group, the repeated measures one-way ANOVA revealed a significant effect of the exercise intervention on SBP (F (1.84, 51.6) = 328, *p* < 0.001) and DBP (F (1.6, 44.9) = 7.94, *p* = 0.002). SBP increased at peak exercise (Tukey’s multiple comparisons test, *p* < 0.001) and decreased below the baseline during recovery (Tukey’s multiple comparisons test, *p* < 0.001). DBP remained unchanged at maximum exercise (Tukey’s multiple comparisons test, *p* = NS) and dropped to values below the baseline during recovery (Tukey’s multiple comparisons test, *p* < 0.001). 

In the high responder group, the Friedman test revealed a significant effect of the exercise intervention on SBP (Friedman statistic: 54.1, *p* < 0.001) and the repeated measures one-way ANOVA revealed a significant effect of the exercise intervention on DBP (F (1.75, 47.3) = 25.8, *p* < 0.001). SBP increased at peak exercise and decreased below the baseline during recovery (Dunn’s multiple comparisons test, *p* < 0.001 for all three comparisons). DBP remained unchanged at maximum exercise (Tukey’s multiple comparisons test, *p* = NS) and dropped to values below the baseline during recovery (Tukey’s multiple comparisons test, *p* < 0.001).

Additionally, the peak HR was significantly lower in the low responder group than in the high responder group (186 ± 6 bpm vs 190 ± 8 bpm, *p* = 0.038).

The baseline apelin-13 level showed a negative correlation with the maximum MET, relative VO_2_max, circulatory power, and circulatory stroke work in the Δapelin-13 low responder group (Figure 3). These associations were absent in the Δapelin-13 high responder group.

In the Δapelin-13 high responder group, a higher Δapelin-13 resulted in a lower power output (Spearman correlation; r = −0.526, *p* = 0.004). Additionally, Δapelin-13 showed a negative correlation with VO_2_max (Spearman correlation; r = -0.397, *p* = 0.036) and VCO_2_max (Spearman correlation; r = −0.446, *p* = 0.017). 

### 2.4. Apelin-36 Response to a Single Bout of Exercise 

The median baseline apelin-36 level was 60.2 pg/mL (IQR, 49.6–79.5 pg/mL; 95% CI, 56.9–105 pg/mL), peak apelin-36 was 150 pg/mL (IQR, 91.6–203 pg/mL; 95% CI, 127–191 pg/mL), and recovery apelin-36 was 45.5 pg/mL (IQR, 36.1–66.6 pg/mL; 95% CI, 43.4–61.3 pg/mL). The Friedman test revealed a significant difference among the three timepoints (Friedman statistic: 30.1, *p* < 0.001), while Dunn’s multiple comparisons test showed a significant change between all three timepoints (peak vs baseline, *p* = 0.010; peak vs recovery, *p* < 0.001; baseline vs recovery, *p* = 0.033). Similar to apelin-13, the general response to the acute load was an increase in apelin-36. Therefore, we analyzed the peptide response on an individual level. Figure 4B shows the peak–baseline values for apelin-36 (Δapelin-36) for all participants in ascending order. The response to the physical load was heterogeneous, with a mean Δapelin-36 level of 78.1 ± 97.4 pg/mL.

Regarding the association between the two apelin fragments, we found negative correlations between apelin-13 levels at all three timepoints and Δapelin-36 (Figure 5).

## 3. Discussion

The present study provided evidence, for the first time, that the plasma levels of apelin-13, on average, show a transient increase in response to maximal cardiorespiratory exercise testing in professional soccer players. Importantly, we detected a considerable heterogeneity in exercise-induced apelin-13 release. Most of the subjects experienced an elevation in their apelin-13 levels; however, a smaller proportion of participants had a reduced level immediately after the physical load. Given the widespread effects of apelin as a cardiokine and myokine, one can anticipate that the diverse apelin response may reflect variations in endurance performance.

Prior studies produced conflicting results regarding the impact of acute exercise on plasma apelin levels. Apelin increased in response to a single bout of cycling sprint interval exercise in healthy individuals [9], and to a swimming test in athletes [8]. In contrast, no change was observed after maximal and submaximal treadmill running bouts in healthy individuals [11]. Moreover, a reduction in apelin concentration was detected immediately after a marathon race [32]. These conflicting results might be explained by the apparent differences in exercise testing modalities and the training status of participants. In addition, the wide interindividual variability observed in our study might explain the existing controversy about the influence of exercise on apelin release. In contrast to apelin-13, NT-proBNP levels remained unchanged. In line with our findings, the short-term maximal exercise test failed to modulate NT-proBNP levels [34], whereas it increased significantly after prolonged strenuous exercise (e.g., marathon running) [35]. These results may suggest that apelin-13 is a more sensitive marker of exercise-induced stress than NT-proBNP.

Although several tissues may contribute to exercise-induced apelin secretion, it appears that the skeletal muscle is a primary source of apelin production during exercise. Skeletal muscle cell contraction stimulates apelin secretion, and the skeletal-muscle-specific knockdown of apelin results in an impaired exercise-induced elevation of plasma apelin levels in mice [30]. Moreover, mice with myofiber-specific overexpression of the Tead1 transcription factor have reduced apelin mRNA and peptide levels in the skeletal muscle in association with lower circulating apelin levels [36]. In addition to myofibers, the vascular endothelium was reported to produce apelin [37]. Since the apelin/apelin receptor system is sensitive to changes in shear stress [38], one may speculate that exercise-induced hyperemia may augment the endothelial secretion of apelin in the vasculature of the working skeletal muscle and coronary circulation. Notably, the observed increases in systemic apelin levels presumably reflect merely the spillover of a tiny fraction of that produced locally.

Intriguingly, pre-exercise baseline SBP and DBP were significantly higher in the Δapelin-13 (i.e., peak–baseline values for apelin-13) high responder group compared to the low responder group. The immediate pre-exercise phase can be described as a period of emotional, behavioral, and physiological excitement about the upcoming challenge. As such, the rise in blood pressure before exercise reflects a cardiovascular adaptation to a stressful situation [39]. In middle-aged normotensive men, elevated SBP (+20.2 ± 14.2 mm Hg (mean ± SD)) and DBP (+9.8 ± 7.7 mm Hg (mean ± SD)) were detected in anticipation of a bicycle ergometer exercise test compared with resting conditions [39]. Since a stronger anticipatory stress response is characterized by the excessive activation of the hypothalamic–pituitary–adrenal axis and the sympathomedullary system [40], the interindividual variability of the Δapelin-13 response may be related to the inherent reactivity of these systems. It should be noted that the overall BP response to exercise testing in high and low responders was consistent with the expected changes described previously in young elite athletes [41]. SBP increased at peak exercise and decreased below baseline during recovery, whereas DBP remained unchanged at maximum exercise and dropped to values below baseline during recovery, probably as a result of extensive peripheral vasodilation. 

During vigorous dynamic exercise, the blood flow of the contracting skeletal muscle can increase nearly 100-fold [12], necessitating a substantial rise in cardiac output. It is well established that a strong relationship exists between maximal cardiac output, VO_2_max, and peak exercise performance [13].

The metabolic equivalent is a practical parameter to describe exercise intensity based on the basic metabolic rate [42]. The soccer players reached a MET at least four times the resting values during the cardiopulmonary exercise test, and a significant positive correlation between Δapelin-13 and maximal MET was found in the entire cohort. The traditional gold standard for assessing aerobic endurance or power has been the VO_2_max test. Relative VO_2_max is an indicator of aerobic capacity and cardiorespiratory fitness [43]. A positive correlation was observed between Δapelin-13 and relative VO_2_max levels.

Circulatory power, the product of VO_2_max by peak systolic arterial pressure, is a non-invasive surrogate of cardiac power and represents an excellent index of the pumping capacity of the heart [44]. In our study, the maximum circulatory power values achieved by the soccer players during the exercise test were higher than the previously reported normal values in healthy, age-matched individuals [45]. Importantly, changes in apelin-13 levels showed a positive correlation with maximal circulatory power. Of note, the correlations between the apelin-13 response (i.e., Δapelin-13) and maximum MET, relative VO_2_max, and circulatory power were mainly driven by players in the Δapelin-13 low responder group.

In the present study, relative VO_2_max showed a strong association with maximal MET and peak circulatory power, reinforcing the intimate relationship between the degree of physical performance, whole-body oxygen consumption, and pumping capability of the heart during maximal cardiorespiratory exercise testing in professional athletes. In a systematic review [46], relative VO_2_max values between 59.2 and 66.6 mL/kg/min were reported for elite male soccer players, which is comparable to the relative VO_2_max values of our sample population. Scarce information is available on plasma apelin levels in soccer players. Sanchis-Gomar and co-workers reported a fluctuation in apelin levels during a season in professional soccer players from an Italian Serie A team; however, no association was found between baseline apelin levels and changes in performance [47]. Our observation that exercise-induced changes in the plasma apelin-13 levels correlate with relative VO_2_max, maximal MET, and peak circulatory power suggest that apelin may contribute to peak athletic performance. Apelin can increase cardiac output by stimulating cardiac contractility [19,24,48,49] and augmenting the Frank–Starling mechanism [19,24,49]. The positive inotropic effect of apelin is nitric-oxide-independent [19], relying on enhanced myofilament Ca^2+^ sensitivity [24,50], and may include the activation of protein kinase Cε, extracellular-signal-regulated kinase 1/2, and myosin light-chain kinase [48], as well as a reduced phosphorylation of cardiac troponin I [24]. An increase in the main determinants of cardiac output, e.g., heart rate, ventricular work, and myocardial contractility, results in a rise in myocardial oxygen demand, which is matched mainly by augmenting coronary blood flow [51]. It has been previously reported that the activation of large conductance, calcium-activated potassium channels (BK_Ca_) by nitric oxide mediates the apelin-induced relaxation of isolated rat coronary arteries, independent of the guanylyl cyclase/cGMP/protein kinase G pathway [25]. Enhanced apelin release may potentiate nitric oxide production by endothelial nitric oxide synthase [27,28], augmenting coronary blood flow in response to exercise. Thus, apelin may balance the oxygen demand and supply in the heart upon physical load. Moreover, apelin may enhance skeletal muscle performance by modulating exercise-induced hyperemia and muscle metabolism via increased glucose utilization [31].

Several lines of evidence suggest a role for endogenous apelin/apelin receptor system in the long-term regulation of cardiac and skeletal muscle homeostasis. In the heart, apelin deficiency triggered premature cardiac aging with declining cardiac function, whereas the systemic restoration of apelin ameliorated these abnormalities [52,53]. The skeletal-muscle-specific knockdown of apelin resulted in reduced muscle mass, muscle weakness, and reduced exercise capacity in aged mice. Apelin supplementation during aging reversed sarcopenia and enhanced muscle function by triggering mitochondriogenesis, autophagy, and anti-inflammatory pathways in myofibers as well as enhancing muscle regeneration by targeting muscle stem cells [30]. Our observation that a single bout of exercise triggers an increase in plasma apelin levels is likely to occur in a repetitive manner during exercise training and may contribute to the long-term exercise-induced adaptation of the cardiac and skeletal muscles; one may anticipate that this improvement may be greater among low Δapelin-13 responders.

In addition to apelin-13, we characterized changes in plasma apelin-36 levels in response to maximal cardiorespiratory exercise testing. Our results show that an acute exercise bout, on average, increased apelin-36 levels; however, a wide interindividual variability was observed in a similar manner to that of apelin-13. Interestingly, Δapelin-36 displayed an inverse relationship with baseline, peak, and recovery apelin-13 levels. Therefore, participants who had a higher apelin-13 level at any timepoints before, during, or after the exercise test had a smaller apelin-36 response. Initially, it was suggested that cleavage took place sequentially with pro-apelin (apelin-55) being first cleaved to apelin-36 and then into the shorter active C-terminal fragments, including apelin-13. Later, however, it was shown that furin can directly cleave pro-apelin to apelin-13 without producing longer isoforms. Importantly, apelin fragments display substantial variation in receptor affinity, signaling, and cellular effects, with shorter apelin isoforms consistently showing a higher biological efficacy [18]. Contrary to our observation that Δapelin-13 correlates with the measures of physical performance and whole-body oxygen consumption, Δapelin-36 did not show a correlation with maximal MET or relative VO_2_max, suggesting that apelin-13 and apelin-36 may play distinct roles during demanding exercise.

In conclusion, our study is the first, to our knowledge, to report changes in the plasma levels of apelin-13 and apelin-36 in response to a single bout of maximum exercise in professional soccer players. Both apelin fragments increased upon extreme physical load; however, the apelin responses showed a striking interindividual variability. Changes in apelin-13 levels correlated with the measures of physical performance, whole-body oxygen consumption, and pumping capability of the heart. These data support the hypothesis that apelin, as an exerkine, may contribute to peak athletic performance. Thereby, the measurement of apelin release may serve as an index of physical effort.

## 4. Materials and Methods

### 4.1. Participants

A total of 58 healthy, normotensive, Hungarian male soccer players (age: 22.9 ± 4.7 years) were included in this study. All participants were of Caucasian origin, self-reported non-smokers, and had no known cardiovascular diseases.

### 4.2. Study Protocol

To determine the changes in apelin concentration upon physical load, the participants underwent a physical stress test carried out in an exercise physiology laboratory (Department of Health Sciences and Sports Medicine, Hungarian Sports University, Budapest, Hungary). A maximum incremental treadmill running test was implemented (2 min warm-up at 8 km/h speed, which was increased to 10 km/h and then remained constant; elevation was 0% in the first 3 min, and then increased 1.5% after each minute). The treadmill test was performed under standard laboratory conditions. The median temperature was 24.7 °C and the relative humidity was 39.5%. The tests were terminated if a subject was unable to continue (volitional exhaustion).

### 4.3. Blood Sampling and Analysis

Standard phlebotomy was performed by qualified personnel before the load (baseline), immediately after the load (peak), and 30 min into the restitution phase (recovery). The plasma samples were centrifuged (4 °C, 1600× *g*, 15 min), and the supernatant sera were collected, frozen in liquid nitrogen as soon as possible, and stored at −80 °C until the measurements were performed. 

A PowerCube gas analyzer unit supplied by Ganshorn (Niederlauer, Germany) was used to measure peak VO_2_ values; the gas analyzer was calibrated before each measurement. An Omron MX2, Cardiosys Human ECG (Experimetria Kft., Budapest, Hungary) was employed for monitoring blood pressure and heart rate. Heart rate and the gas exchange parameters were registered continuously during physical stress. SBP and DBP were recorded at three time points during the exercise test (baseline, peak, and recovery). Baseline BP was measured in the standing position on the treadmill before initiating the exercise test. Lactate concentrations were recorded at three time points (baseline, peak, and recovery) and measurements were performed on a Biosen C-line Glucose and Lactate Analyzer (Frank Diagnosztika Kft., Budapest, Hungary). 

The peptide ELISA analysis was conducted at the Faculty of Health Sciences, University of Pécs, Pécs, Hungary. Circulating peptide (apelin-13, apelin-36, and NT-proBNP) concentrations were measured using a Multiskan FC Microplate Photometer (Thermo Fisher Scientific; Waltham, MA, USA). Apelin-13 and apelin-36 (Cusabio; Houston, TX, USA) were measured by a quantitative sandwich assay technique in duplicate. Intra-assay precision was <8% and <15% and inter-assay precision was <10% and <15%, respectively. No significant cross-reactivity or interference between human AP-13/AP-36 and analogues was observed. NT-proBNP (Biomedica; Vienna, Austria) was measured by a sandwich immunoassay in duplicate. Intra-assay precision was ≤4% and inter-assay precision was ≤7%.

### 4.4. Cardiopulmonary Exercise Parameters

In addition to blood pressure, heart rate, and blood lactate, other parameters influenced by the exercise test were measured or calculated. The measured parameters included metabolic equivalent (MET; 1 MET = 3.5 mL O_2_/kg body weight/minute), peak power output, VO_2_max and relative VO_2_max (parameters of maximum O_2_ consumption), maximum CO_2_ production (VCO_2_), maximum ventilation (VE), and maximum rate of respiration (number of breath/minute). The calculated parameters included baseline rate pressure product (baseline RPP = baseline systolic BP × baseline HR), peak rate pressure product (peak RPP = peak systolic BP × peak HR), rate pressure product reserve (RPP reserve = peak RPP-baseline RPP), maximum respiratory quotient (RQ = maximum VCO_2_/VO_2_max), VE/VO_2_, VE/VCO_2_, circulatory power (VO_2_max × SBP), and circulatory stroke work (circulatory power/peak HR).

The following criteria were used to confirm extreme physical load: (1) duration of the activity should be at least 8 min; (2) maximum HR ≥ 160–180 beats per minute, depending on the age of the participants; (3) RQ value ≥ 1.1 at the peak of the load; (4) when increasing the load, the oxygen consumption should reach its maximum; and (5) the lactate concentration at the maximum load should be 8 mmol/L or higher [54].

### 4.5. Ethics

The study was approved by the National Public Health Center of Hungary (15117–9/2018/EÜIG, 24 May 2018). All subjects provided written informed consent prior to participation in the study. The study was conducted in accordance with the World Medical Association Declaration of Helsinki.

### 4.6. Statistical Analysis

For the statistical analysis, GraphPad Prism (version 7.0.0 and 9.4.1, GraphPad Software, Boston, MA, USA) and Microsoft Excel 2016 (Microsoft Corporation, Redmond, WA, USA) were used. The Gaussian distribution was tested using the D’Agostino–Pearson omnibus normality test. The results are presented as mean ± standard deviation (SD) for continuous normally distributed data and as median and interquartile range (IQR) for continuous non-normally distributed data. Temporal changes in the normally distributed data (i.e., plasma apelin-13, DBP in the whole cohort and in both subgroups, and SBP in the low responder group) due to acute exercise intervention were evaluated by a repeated measures one-way ANOVA test with time as a within-subject factor (baseline, peak, and recovery). To protect against the violation of the sphericity assumption, the Geisser–Greenhouse correction was used. When the main effect was statistically significant, a Tukey’s multiple comparisons post hoc test was performed for pairwise comparisons. Temporal changes in non-normally distributed data (i.e., plasma apelin-36 and NT-proBNP, SBP in the whole cohort and in the high responder group, heart rate, and blood lactate levels) in response to acute exercise intervention at 3 different timepoints (baseline, peak, and recovery) were analyzed using a non-parametric Friedman test. Where appropriate, a Dunn’s multiple comparisons post hoc test was performed for pairwise comparisons. An unpaired Student’s *t*-test was used to compare the normally distributed values between the 2 groups (i.e., baseline SBP, peak SBP, baseline DBP, peak DBP, recovery DBP, peak HR, baseline RPP, peak RPP, recovery RPP, run time, maximum MET, power output, VO_2_max, maximum RQ, maximum breathing/minute, VE/VO_2_, VE/VCO_2_, and peak lactate), while a Mann–Whitney test was used to compare the non-normally distributed values between the 2 groups (i.e., recovery SBP, baseline HR, recovery HR, relative VO_2_max, maximum VCO_2_, maximum VE, circulatory power, circulatory stroke work, baseline lactate, and recovery lactate). Likewise, the correlation was analyzed using either a Pearson correlation for the normally distributed data or a Spearman correlation for the data that did not pass the normality test. The applied statistical tests are detailed in each figure legend. Differences were considered statistically significant at *p* < 0.05.

## Figures and Tables

**Figure 1 ijms-24-08195-f001:**
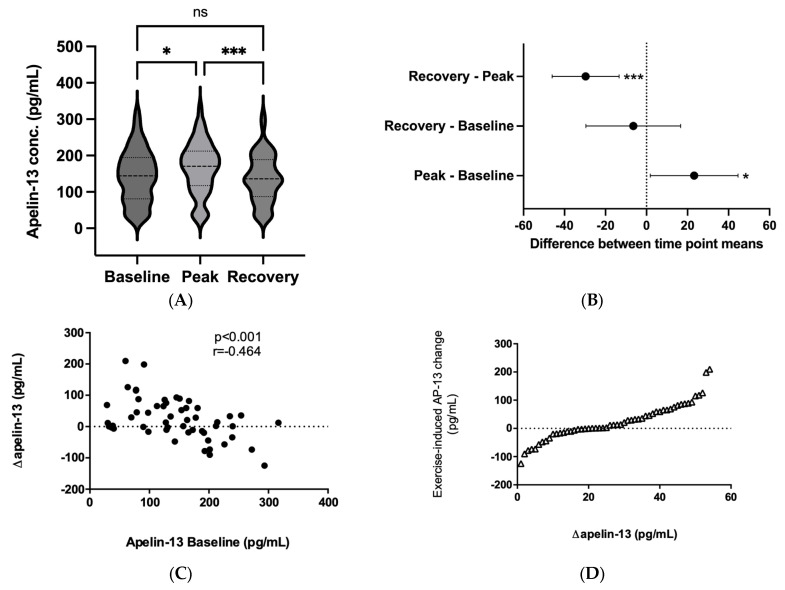
(**A**) Violin plots comparing the plasma levels of apelin-13 before (baseline), immediately after (peak), and 30 min after (recovery) the vita maxima treadmill test. Medians and the 75th and 25th percentiles are shown within the violin plots. Data were analyzed by a repeated measures one-way ANOVA followed by Tukey’s multiple comparison test (n = 54), ns: no statistical significance, * *p* < 0.05, ****p* < 0.001. (**B**) Difference between time point means (baseline, peak, and recovery) of apelin-13 levels with multiplicity-adjusted 95% confidence intervals according to Tukey’s test, **p* < 0.05, ****p* < 0.001. (**C**) Spearman correlation of baseline apelin-13 and Δapelin-13 (N = 58). (**D**) Individual apelin-13 responses to the exercise test. Each point represents the change in a subject’s apelin-13 level from baseline to maximum load. Baseline values are subtracted from peak values and sorted in ascending order (N = 58).

**Figure 2 ijms-24-08195-f002:**
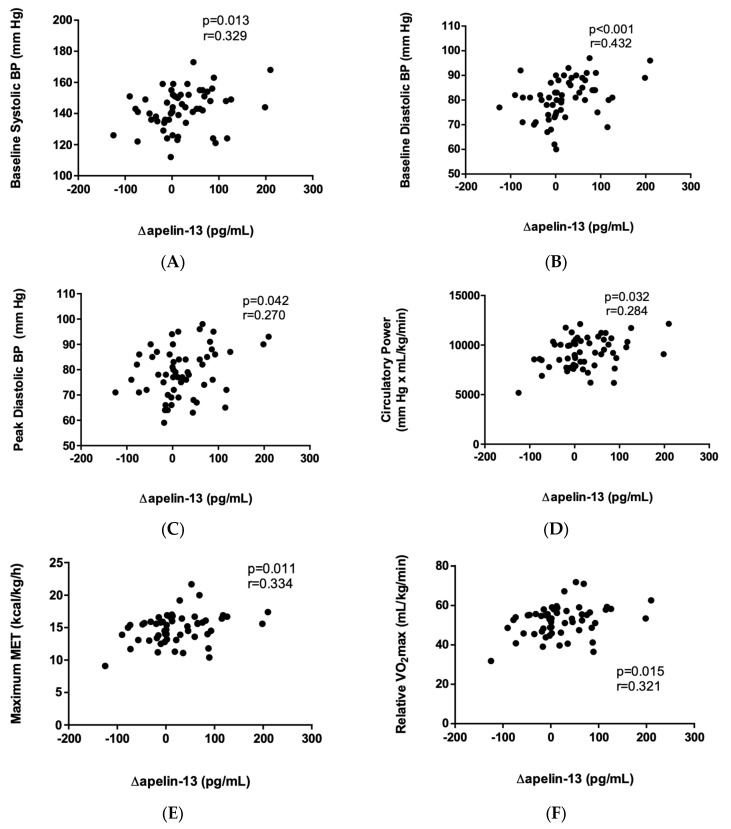
Correlation of Δapelin-13 and baseline systolic BP (**A**), baseline diastolic BP (**B**), peak diastolic BP (**C**), circulatory power (**D**), maximum MET (**E**), and relative VO_2_max (**F**) (N = 58). Data were analyzed by Spearman correlation.

**Figure 3 ijms-24-08195-f003:**
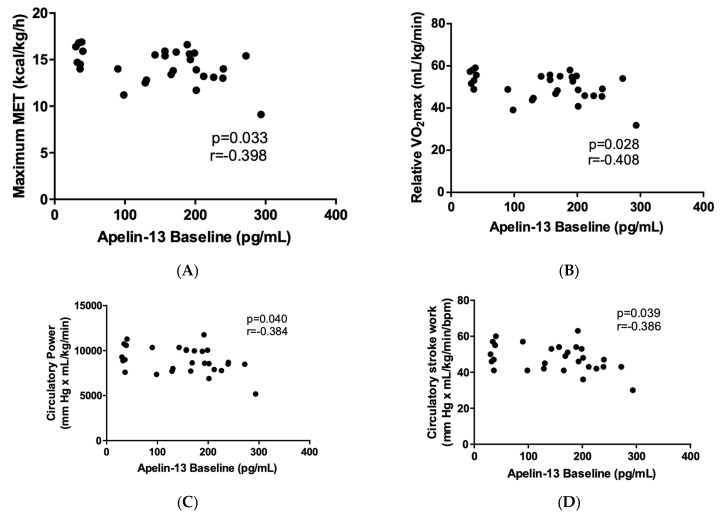
Correlation of apelin-13 baseline levels with the maximum MET (**A**), relative VO_2_max (**B**), circulatory power (**C**), and circulatory stroke work (**D**) in the Δapelin-13 low responder group (N = 29). Data were analyzed by Pearson correlation.

**Figure 4 ijms-24-08195-f004:**
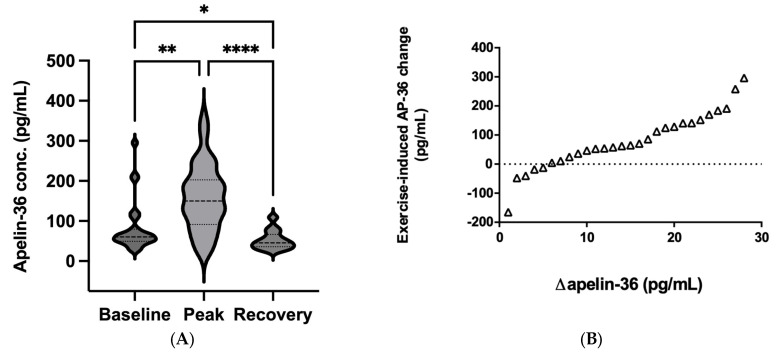
(**A**) Violin plots comparing the plasma levels of apelin-36 before (baseline), immediately after (peak), and 30 min after (recovery) the vita maxima treadmill test. Medians and the 75th and 25th percentiles are shown within the violin. Data were analyzed by a Friedman test followed by a Dunn’s multiple comparisons test (N = 28), * *p* < 0.05, ** *p* < 0.01, **** *p* < 0.001. (**B**) Individual apelin-36 responses to the exercise test. Each point represents the change in a subject’s apelin-36 level from baseline to maximum load. Baseline values are subtracted from peak values and sorted in ascending order (N = 28).

**Figure 5 ijms-24-08195-f005:**
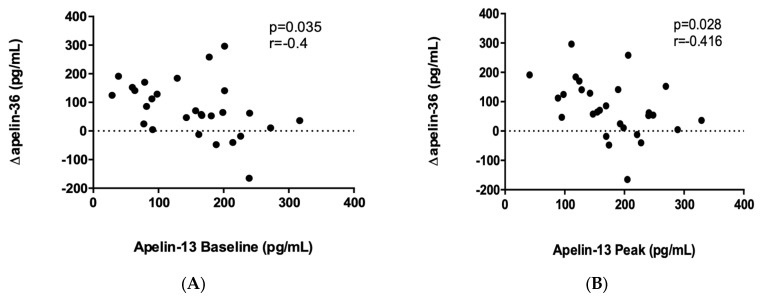
Correlation between baseline apelin-13 and Δapelin-36 (**A**), peak apelin-13 and Δapelin-36 (**B**), and recovery apelin-13 and Δapelin-36 (**C**). Data were analyzed by Pearson correlation (**A**,**B**) or Spearman correlation (**C**) based on the distribution of the data (N = 28).

**Table 1 ijms-24-08195-t001:** Cardiovascular, cardiorespiratory, and metabolic parameters of the athletes at baseline, peak, and recovery timepoints.

	All Participants (N = 58)	ΔApelin-13		*p*-Value
		Low Responders (n = 29)	High Responders (n = 29)	
Baseline SBP (mm Hg) *	143 ± 13	139 ± 12	147 ± 13	** *p* ** ** = 0.048**
Peak SBP (mm Hg) *	177 ± 16	178 ± 15	176 ± 19	*p* = 0.658
Recovery SBP (mm Hg) †	127 (120–132)	127 (120–132)	128 (120–133)	*p* = 0.715
Baseline DBP (mm Hg) *	81 ± 8	77 ± 8	85 ± 7	** *p* ** ** = 0.001**
Peak DBP (mm Hg) *	79 ± 9	77 ± 9	81 ± 10	*p* = 0.071
Recovery DBP (mm Hg) *	72 ± 7	72 ± 8	73 ± 8	*p* = 0.557
Baseline HR (bpm) †	71 (61–80)	68 (61–77)	73 (63–80)	*p* = 0.512
Peak HR (bpm) *	189 ± 8	186 ± 6	190 ± 8	** *p* ** ** = 0.038**
Recovery HR (bpm) †	86 (77–93)	88 (77–91)	86 (79–95	*p* = 0.560
Baseline RPP (mm Hg × bpm) *	10,143 ± 2134	9637 ± 2071	10,506 ± 1749	*p* = 0.135
Peak RPP (mm Hg × bpm) *	33,409 ± 3132	33,209 ± 3122	33,326 ± 3411	*p* = 0.876
RPP Reserve (mm Hg × bpm) *	23,266 ± 3001	23,572 ± 3254	22,820 ± 3034	*p* = 0.540
Run time (min) *	11.2 ± 1.5	10.9 ± 1.7	11.2 ± 1.4	*p* = 0.285
Max MET *	14.9 ± 2.2	14.3 ± 1.8	15.5 ± 2.6	*p* = 0.084
Power output (Watt) *	395 ± 43	394 ± 46	392 ± 43	*p* = 0.950
VO_2_max (L/min) *	3.97 ± 0.57	3.85 ± 0.48	4.13 ± 0.67	*p* = 0.132
Relative VO_2max_ (mL/kg/min) †	52.7 (47–57.3)	51.6 (45.9–55.2)	55.4 (49.3–59)	*p* = 0.069
Max VCO_2_ (L/min) †	5.25 (5–5.78)	5.21 (4.65–5.74)	5.27 (5.17–5.9)	*p* = 0.313
Max RQ *	1.37 ± 0.10	1.38 ± 0.09	1.35 ± 0.09	*p* = 0.346
Max VE (L/min) †	150 (136–162)	142 (128–156)	151 (140–163)	*p* = 0.179
Max breathing/minute (breath/min) *	53.4 ± 7.5	51.8 ± 7.5	53.2 ± 7.6	*p* = 0.470
VE/VO_2_*	37.8 ± 6.2	36.9 ± 5.7	37.2 ± 6.3	*p* = 0.889
VE/VCO_2_*	28.0 ± 3.3	27.4 ± 3.4	27.9 ± 3.3	*p* = 0.597
Circulatory power (mm Hg × mL/kg/min) †	9141 (8054–10,335)	8691 (7841–10,070)	9661 (8317–10,734)	*p* = 0.220
Circulatory stroke work (mm Hg × mL/kg/min/bpm) †	49 (43–54.8)	47 (42.5–54)	51 (42.5–56.8)	*p* = 0.343
Baseline lactate (mmol/L) †	0.9 (0.7–1.2)	0.9 (0.75–1.2)	0.9 (0.73–1.3)	*p* = 0.526
Peak lactate (mmol/L) *	11.10 ± 2.21	10.90 ± 2.13	11.10 ± 2.44	*p* = 0.977
Recovery lactate (mmol/L) †	4.52 (3.22–5.61)	4.4 (3.41–6.21)	4.04 (2.96–5.78)	*p* = 0.243

Variables are expressed as mean ± SD or median (interquartile range, IQR: 25th and 75th percentiles). Data were analyzed by an unpaired Student’s *t*-test (*) or Mann–Whitney test (†) to compare the parametric and non-parametric values between the two groups, respectively. Abbreviations: DBP, diastolic blood pressure; HR, heart rate; MET, metabolic equivalent; RPP, rate pressure product; RQ, respiratory quotient; SBP, systolic blood pressure; VCO_2_max, maximum carbon dioxide production; VE, ventilation; VO_2_max, maximum oxygen consumption.

## Data Availability

All data presented in this study are contained within the article.

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
