# Peer review of "Apelin as a Potential Regulator of Peak Athletic Performance"

_ijms, 2023, doi:10.3390/ijms24098195_

Round 1

Reviewer 1 Report

The authors have done a great job. The topic is well contextualised, and the results are well presented. However, the authors focus the discourse on athletes, but the sample used is very specific (soccer players). Therefore, this detail should be well specified throughout the manuscript to avoid confusion. In addition, there are some points in the statistical analysis that need to be specified. Below are some comments that should be clarified to improve their understanding:

The text of the manuscript focuses on the performance of the athletes, but the sample is very specific. In this case, soccer players. Therefore, the text should be rewritten to specify the type of athletes and not generalise it to all athletes, because athletes in each sport have specific physical characteristics. Information about the athletes in this sport should also be added in the introduction and the results should be compared with other athletes in the same sport in the discussion.

Statistical correlations:

Lines 111-112: Why is the first value shown with mean and SD and the second with median and interquartile range?

Lines 113-114: On the one hand, from the wording it seems that correlation has been done by means of a mean and a median, is this the case? If not, it should be explained what centrality data has been used in this analysis. On the other hand, what test has been carried out to obtain this correlation?

Line 401: Indicate which analysis has been used for each statistical test.

Lines 368: indicate how the METs have been calculated.

Line 98 and statistical analysis section: Why do the authors use a different post-hoc test for each variable (Turkey and Dunn)? Also, why have these post-hocs been chosen? It is known that Dunn is a post-hoc that is not very restrictive, and it is easier to obtain significant differences. Therefore, it is advisable to use a more restrictive one, such as Bonferroni.

Statistical analysis section: Specify by what criteria it was decided to use parametric and non-parametric tests. In this section it is indicated that parametric tests are used for statistical analyses, but in the results some values are shown with median and interquartile range. Furthermore, in lines 89-91 of the Results section, it is stated that the Student's t-test or Mann-Whitney test is used, but the authors do not specify which variables are used for each test.

Reviewer 2 Report

Dear Authors,

It is good to see this peer-review article from your hard-working. Here are some of my thoughts.

Comment 1 first of all, please check figure 1, figure 2, figure 5 in this PDF file, it looks like something went wrong when the authors save their manuscript from word file to PDF file.

Comment 2 Line 23, ‘Our study is the first to provide evidence that plasma levels of apelin-13 and apelin-36 show a transient increase in response to maximal exercise...’ It seems like this is not your conclusion. Please focus on what are the facts you can generate from your data.

Comment 3 Line 24, there may be something following the word ‘response’. The authors should specify what they indicate the apelin-13 response to. I assume the authors want to say ‘apelin-13 response to physical performance‘, rather than the ‘measurement’ of physical performance. Please re-arrange this sentence if necessary.

Comment 4 Line 43-70, the underlying mechanism of apelin-13 response to exercise associate with its vasodilation role through ACE, PKC, NOS, G-protein-coupled apelin receptor signaling pathways. In background section, authors majorly mentioned apelin-13 as cardiokine, myokine, and adipokine, while, I would recommend the authors to pay more attention to their underlying mechanism and signaling pathways.

Comment 5 Table1. Like we both acknowledged, apelin-13 can play a vasodilation role, and for those high responders with greater level of apelin-13, baseline SBP might be lower than low responders since they are all healthy adults, while, table 1 row 1 went opposite. And Table 1 also showed that mean SBP of High responders was 147mm/Hg, but we know that SBP of healthy adult should be less than 140 mm/Hg. Are you sure this group was recruited from healthy adults? Please check the original datasets.

Comment 6 Line 164-183, please refer to the comment 5. Baseline apelin-13 showed weak to moderate negative correlation with exercise capacity among low responders and high responders with some of the exercise capacity indexes. Please confirm if this is consistent with the conclusion that greater apelin-13 associate with higher performance.

Comment 7 Since apelin-13 increase during exercise, does low responders seems to benefit more from exercise training in current findings? 

Comment 8, Line 280, please check the mixed use of parentheses.

Comment 9, Apelin-13 can improve NO synthesis from vascular endothelial cells and decrease Ang-II in the peripheral circulation. Ang-II and NOS are well recognized for regulating blood pressure (NO can lead to vasodilation and Ang-II lead to vasoconstriction), It is suggested that the authors should to measure and describe Ang-II, NOS, etc. along with their experiment and their association with Apelin-13 and Apelin-36. Thus, we can explore some issues from the mechanism.

Best regards,

Reviewer

Round 2

Reviewer 1 Report

The authors have answered the questions correctly and have also implemented substantial changes to the manuscript.

Author Response

We wanted to say thank you to the Reviewer for the valuable insights to improve our manuscript.

Reviewer 2 Report

Dear Authors,

The manuscript improved than the last version. It seems like you are not answering each of our comments. I would suggest that you may provide your answers to all our comments if possible. The following are the comments you did not response.

Previous comment 4, the underlying mechanism of apelin-13 response to exercise associate with its vasodilation role through ACE, PKC, NOS, G-protein-coupled apelin receptor signaling pathways. In background section, authors majorly mentioned apelin-13 as cardiokine, myokine, and adipokine, while, I would recommend the authors to pay more attention to their underlying mechanism and signaling pathways.

Comment 5 Table1. Like we both acknowledged, apelin-13 can play a vasodilation role, and for those high responders with greater level of apelin-13, baseline SBP might be lower than low responders since they are all healthy adults, while, table 1 row 1 went opposite. And SBP of healthy adult should be less than 140 mm/Hg. Please check the orignal datasets.

In addition to comment 5, it seems the authors added quite a lot significance notes(asterisk *, and cross ), while, the p-values are actually greater than 0.05. Please check the original data sets.

Comment 6 Line 164-183, please refer to the comment 5. Baseline apelin-13 showed weak to moderate negative correlation with exercise capacity among low responders and high responders with some of the exercise capacity indexes. Please confirm if this is consistent with the conclusion that greater apelin-13 associate with higher performance.

Comment 7 Since apelin-13 increase during exercise, does low responders seems to benefit more from exercise training in current findings? 

Comment 9, Apelin-13 can improve NO synthesis from vascular endothelial cells and decrease Ang-II in the peripheral circulation. Ang-II and NOS are well recognized for regulating blood pressure (NO can lead to vasodilation and Ang-II lead to vasoconstriction), we suggest the authors to measure and describe Ang-II, NOS, etc. along with their experiment and their association with Apelin-13 and Apelin-36.

Best,

Reviewer

Round 3

Reviewer 2 Report

Dear Authors,

It is great to see the improvement you have made so far, and from your last revision, I saw a lot of improvements.

We agree with you that circulating apelin-13 and apelin-36 are sensitive to exercise training, and their levels are individualized among different people. However, the underlying cellular and molecular mechanism is not the research purpose of your current study.

Though there are several concerns I have left, I can recommend this version of manuscript to be published.

Best regards,

Reviewer